# Microstructural Characteristics of Cellulosic Fiber-Reinforced Cement Composite

**DOI:** 10.3390/ma18010059

**Published:** 2024-12-27

**Authors:** Jae-Yoon Han, Young-Cheol Choi

**Affiliations:** Department of Civil and Environmental Engineering, Gachon University, Seongnam 13120, Gyeonggi-do, Republic of Korea; wodbs1763@gachon.ac.kr

**Keywords:** natural fiber, microstructure, compressive strength, internal curing, hydration

## Abstract

The microstructural evolution and hydration behaviors of cement composites incorporating three natural fibers (abaca, hemp, and jute) were investigated in this study. Mercury intrusion porosimetry was used to assess the microstructural changes, focusing on the pore-size distribution and total porosity. Additionally, the hydration characteristics were analyzed using setting time measurements and isothermal calorimetry to track the heat flow and reaction kinetics during cement hydration. Although the fibers tended to delay the initial stages of cement hydration, their internal curing effect ultimately led to a higher long-term compressive strength and a denser microstructure. Consequently, the use of these natural fibers in cement composites can enhance their durability and promote sustainable construction materials.

## 1. Introduction

Recently, the construction industry has focused on developing new eco-friendly materials, owing to the need for sustainable and environmentally friendly resources [1]. Concrete is the most widely used construction material globally. However, it is inherently weak in terms of its tensile strength. Cracks in concrete structures are often caused by its low tensile strength, which accelerates the penetration of harmful ions into concrete and ultimately reduces the service life of concrete structures. To address these shortcomings, fibers have been extensively used, with synthetic fibers being particularly effective at enhancing several mechanical properties such as fatigue and tensile strength [2,3,4]. However, synthetic fibers are expensive and often have a high carbon footprint, creating a demand for alternative materials. Natural fibers, which use renewable resources, have emerged as promising substitutes to mitigate the environmental impact of synthetic fibers [5]. Natural fibers, such as kenaf, bamboo, abaca, jute, bagasse, and banana leaves, offer significant advantages over synthetic fibers in terms of cost, sustainability, and availability. In addition, natural fibers possess excellent mechanical properties, making them suitable for applications in cement composites. Their biodegradable nature also simplifies their disposal as construction waste [6]. Consequently, there have been many studies conducted that have investigated the use of natural fibers as reinforcement in cement composites [7,8,9,10,11].

Natural fibers, characterized by low density and high tensile strength, improve the mechanical performance of cement matrices [12]. Some studies have shown that mortars containing natural fibers, such as coconut, jute, and kelp, exhibit increased strength and durability [13]. Awwad et al. [14], however, reported that incorporating 0.5 vol% hemp fiber in concrete reduced the compressive and flexural strength by 19% and 31%, respectively. Kaplan and Bayraktar [15] observed that adding a 10 mm long hemp fiber at 2 vol% yielded the best mechanical performance in cement and mortar specimens. Conversely, Elsaid et al. [16] found that incorporating kenaf fibers in concrete resulted in reduced compressive and flexural strengths compared to control specimens without fibers, while the splitting tensile strength showed a slight improvement. According to Ziane et al. [17], exceeding a certain threshold of natural fiber content in cement matrices leads to decreased mechanical properties, owing to the heterogeneity introduced by the fibers. Abdalla et al. [18] investigated the effects of luffa fibers on the mechanical properties and microstructure of concrete and reported that the addition of 1% luffa fiber led to a 7.6% increase in the compressive strength after 28 d. This enhancement was attributed to the internal curing effect of the luffa fibers, which promoted the continuous formation of hydration products. Natural fibers generally possess a porous lumen structure, which gives them significantly better moisture-absorption properties than synthetic fibers. This high-absorption characteristic, combined with the chemical composition of natural fibers, significantly influences the hydration reaction of cement. Natural fibers typically contain cellulose, hemicellulose, lignin, and pectin, all of which affect the cement hydration process [19,20,21,22,23]. Furthermore, it has been reported that pectin and sugar components present in natural fibers delay cement hydration [24,25].

During the initial mixing of mortar or concrete, the moisture contained in the natural fibers moves from the fibers into the cement matrix as hydration progresses. This results in an internal curing effect, in which moisture is supplied from within the natural fibers. This internal curing effect of natural fibers influences the moisture distribution within the cement matrix during the hydration process, and several studies have been conducted that used this characteristic to reduce shrinkage in cement and concrete [26,27,28,29,30]. It was found that natural fibers acted as reservoirs within the cement matrix, reducing shrinkage. They compensated for moisture loss owing to self-desiccation and showed excellent performance in reducing autogenous shrinkage. Furthermore, the additional moisture supplied to the cement matrix affected the formation and distribution of hydration products, as well as the development of the pore structure. Ultimately, this impacted the mechanical properties and durability of natural-fiber cement composites, indicating the need for in-depth research into their microstructural characteristics. However, there have been few studies on the impact of natural fibers on the microstructure compared with their effects on cement hydration and shrinkage reduction.

In this study, the effects of natural fibers on the microstructure of a cement matrix were investigated. To achieve this, cement paste and mortar specimens were prepared using different types of natural fibers (abaca, hemp, and jute fibers), and performance evaluations were conducted on the hydration characteristics, mechanical properties, and micropore structure. The hydration characteristics were assessed using setting time tests and heat-of-hydration measurements. The mechanical properties were evaluated by measuring the compressive strengths of the mortar specimens at different curing ages. The micropore structure of the natural fiber-reinforced cement paste was analyzed using mercury intrusion porosimetry (MIP) to investigate the pore structure changes over time. The findings of this study are expected to expand the potential applications of different natural fibers and contribute to the development of customized composites tailored to regional characteristics.

## 2. Experimental Details

### 2.1. Materials

In this study, ordinary Portland cement (OPC), a commercially available product produced by Company A in Seoul, Republic of Korea, was used as a binder for natural-fiber cement composites. Table 1 shows the chemical composition of the OPC, as determined by X-ray fluorescence analysis. The density and specific surface area of the OPC were 3.12 g/cm^3^ and 3850 cm^2^/g, respectively. Three types of natural fibers, abaca, hemp, and jute, were used, as shown in Figure 1. The physical properties of the natural fibers are listed in Table 2. Standard sand specified in ISO 679 [31] was used to prepare the mortar specimens. Standard sand is a naturally occurring siliceous material comprising rounded particles and >98% silicon dioxide (SiO_2_). Figure 2 shows the particle size distribution of the OPC, which was measured using a Beckman Coulter LS 230 laser diffraction particle-size analyzer (Beckman Coulter, Brea, CA, USA). The average particle size was 17.2 μm. To prevent hydration reactions during measurements, the OPC was dispersed in an isopropyl alcohol solution.

Abaca is a perennial plant belonging to the Musaceae family that reaches a height of approximately 6 m. The leaves are large, elliptical, and attached to the main stem of the plant, and the lower part of the leaf forms a sheath that wraps around the stem. This sheath contains unrefined fibers with lengths ranging from approximately 1.5 to 3.5 m. The main components of these fibers are cellulose, lignin, and pectin. Hemp is a variety of Cannabis sativa that is cultivated for both consumption and industrial use. Along with bamboo, hemp is one of the fastest-growing plants on Earth, and was one of the first plants to be spun into fibers nearly 50,000 years ago. Jute is a long, rough, and shiny bast fiber produced by the flowering plants of the genus Corchorus, which belongs to the Malvaceae family. Jute fibers mainly consist of cellulose and lignin, and the stem grows to approximately 2–5 m with a basal diameter of approximately 20 mm. The densities of abaca, hemp, and jute fibers are 1.51 g/cm^3^, 1.50 g/cm^3^, and 1.44 g/cm^3^, respectively. All three natural fibers were produced without chemical or thermal treatment and were immersed in water, harvested, and dried.

The dimensions of the three natural fibers were measured using an optical microscope. The average length and diameter of abaca, hemp, and jute fibers were 4.41 mm and 0.31 mm, 5.22 mm and 0.25 mm, and 6.84 mm and 0.13 mm, respectively. Figure 3 shows scanning electron microscopy (SEM) images of the lateral view and cross-section of the three types of natural fibers used in this study. The figure shows that the surfaces of all fibers contained impurities, such as hemicellulose, lignin, pectin, and wax. The cross-sections of all three natural fibers exhibit hollow tubular bundle structures that varied, depending on the type of natural fiber. The average pore diameters in the cross sections of the abaca, hemp, and jute fibers were 9.73 µm, 6.68 µm, and 2.08 µm, respectively. This tubular structure can serve as both a pathway and a reservoir for moisture, contributing to the high water-absorption capability of natural fibers, and making them suitable for use as internal curing agents in cement composites.

Figure 4 shows the thermogravimetric (TG) and derivative TG (DTG) analyses for the natural fibers. The natural fibers used in this study primarily decomposed in the range from 250 to 400 °C. From the DTG results in Figure 4, a peak was observed at approximately 55–70 °C, which is attributed to evaporation or dehydration of water molecules within the natural fibers [35]. Subsequently, three major peaks were observed, with varying sizes. The approximate temperature ranges of these peaks are indicated by dashed lines in Figure 4. The three peaks were attributed to the decomposition of hemicellulose, cellulose, and, although less distinct, the decomposition of lignin [36], respectively. The approximate chemical compositions of the abaca, hemp, and jute fibers are listed in Table 3.

### 2.2. Mixture Proportions

In this study, cement pastes and mortars were prepared to evaluate the properties of cement composites incorporating natural fibers. As shown in Figure 3, the natural fibers have a porous structure, resulting in significant moisture absorption. Considering this absorption, the mix proportions of the cement composites were prepared as shown in Table 4. Here, the term Plain refers to the control specimen without natural fibers, while NFA, NFH, and NFJ are the specimens containing abaca, hemp, and jute fibers, respectively. The water absorptions of the abaca, hemp, and jute fibers were 292%, 214%, and 271%, respectively, and additional water was included in the mix design. A polycarboxylate-based chemical admixture was combined at a constant rate of 8.74 kg/m^3^ in all mixes, to ensure workability. Paste specimens were prepared to investigate their hydration characteristics, and mortar specimens were produced to measure the compressive strength. The standard sand for the mortar specimens was combined at the rate of 1344 kg/m^3^ for all specimens. The materials were first mixed in a Hobart mixer by combining the cement with standard sand at approximately 60 rpm for 30 s. Water, including an additional amount adjusted for the absorption rate of the natural fibers, was then added, and the mixture was blended at approximately 100 rpm for 60 s. To enhance fiber dispersion, half of the fibers were added and mixed at 200 rpm for 15 s, followed by the addition of the remaining half, which was mixed for 5 min.

### 2.3. Test Methods

The setting times of the cement composites incorporating natural fibers were measured in accordance with ISO 9597 [38], using an automatic-setting time tester. A PA8 automatic tester (ACMEL LABO, Saint Pierre du Perray, France) was used to automatically measure the penetration depth over time for the cement paste prepared according to Table 4, and the setting time was determined accordingly. As shown in Figure 3, natural fibers contain numerous pores, resulting in a high moisture-absorption rate. Therefore, to apply natural fibers to cement-based products, it is necessary to consider this characteristic, and it is crucial to accurately measure the moisture absorption rate of natural fibers [13,26,39]. In this study, the vacuum filtration method used in previous studies was used to measure the moisture-absorption rate of the natural fibers [40]. The natural fibers were fully saturated by immersion in water for a sufficient period, and the surface water was removed using a vacuum pump and membrane filter. The moisture-absorption rate was determined by measuring the changes in weight.

To investigate the effect of natural fibers on the early hydration reaction of cement, the heat of hydration of cement composites was measured using an isothermal calorimeter, TAM Air (TA Instruments, New Castle, DE, USA). Approximately 4 g of the cement paste prepared according to Table 4 was placed in a glass vial for measurement, and the heat evolution was automatically recorded. The isothermal calorimeter was set to a curing temperature 24 h prior to the experiment, and maintained at the same temperature throughout the experiment.

To examine the effect of natural fibers on the mechanical properties of cement, mortar specimens were prepared and the compressive strength was measured at different ages in accordance with ISO 679. The mortar specimens were cast in prismatic molds with dimensions of 40 mm × 40 mm × 160 mm and cured in a temperature-controlled chamber at 20 ± 1 °C and relative humidity > 90% for 24 h. After demolding, the mortar specimens were submerged in a water bath maintained at 20 ± 1 °C for further curing. To measure the compressive strength, the mortar specimens were first split into two halves using a splitting test, and the compressive strength was measured for each half. The average compressive strengths of six specimens were used for each variable.

MIP was performed to investigate the effect of natural fibers on the microstructure of the cement composites using an AutoPore IV 9500 (Micromeritics, Norcross, GA, USA) at a maximum pressure of 30,000 psi. The MIP results presented herein are the averages of three individual measurements.

## 3. Results and Discussion

### 3.1. Setting Behavior of Specimens

In this study, the effect of natural fibers on the setting of cement was investigated using an automatic-setting time tester to measure the initial and final setting times of cement paste incorporating natural fibers. The results are presented in Figure 5. The figure shows that the initial and final setting times varied, depending on the fiber type. For the Plain specimen, the initial setting time was 12.3 h and the final setting time was 15.2 h. The inclusion of natural fibers delayed the setting time, with particularly pronounced delays in the final setting time. Among the specimens containing natural fibers, the NFH had the longest initial and final setting times of 15.7 and 19.77 h, respectively, indicating the most significant delay effect. Specimens incorporating abaca and jute fibers also showed delayed setting times compared to the control; however, the extent of the delay was less pronounced than that observed for NFH. This suggests that the incorporation of natural fibers affects the hydration reaction and setting time of cement paste, with the effect varying, depending on the type of fiber used.

Generally, natural fibers, including abaca, hemp, and jute fibers, contain cellulose, hemicellulose, lignin, and pectin. These components can leach out in the high-pH environment of the cement pore solution, potentially delaying cement hydration [21,22,23,24]. Cellulose is composed of glycosidic linkages; when these linkages are hydrolyzed, glucose remains. In the alkaline environment inside the cement matrix, which is created by the hydration of the C_3_A and C_3_S components, glucose undergoes alkaline degradation, resulting in the formation of insoluble salts that coat the surface of the cement clinker, thus inhibiting the hydration reaction. Furthermore, glucose undergoes ring opening in the presence of OH- ions and forms insoluble metal–organic complexes by binding with ions such as Ca^2+^, Al^3+^, and Fe^2+^, which also coat the surface of the cement clinker and inhibit the hydration process [41].

### 3.2. Heat of Hydration of Specimens

Figure 6 shows the specific heat rate and cumulative heat release of the cement-paste specimens containing natural fibers. Figure 6a shows that, for the specific heat-rate results over time, the occurrence time of the second peak increased for specimens with natural fibers, indicating that natural fibers delayed the overall hydration of cement. While the acceleration periods for Plain, NFA, and NFJ were similar, the acceleration period for NFH was significantly delayed, which was consistent with the initial setting-time results. The second peak for the Plain specimen occurred at 16.7 h, whereas NFA and NFJ showed second peaks at 18.0 and 19.4 h, respectively. The second peak for NFH occurred at 22.2 h, which was 1.3× later than that of the Plain specimen. Furthermore, the height of the second peak was significantly reduced for specimens containing natural fibers. The second peak for the NFH was 2.91 mW/g, which was approximately 40% of that of the Plain specimen. These results are consistent with those reported previously. Kochova et al. [42] measured the specific heat rate of paste specimens containing 99.5% pure glucose and hemp fibers. They reported that the second peak occurred two days later for specimens with 0.3% glucose, and the second peak for specimens with hemp fibers occurred one hour later than that of the specimens without hemp fibers. The delay in the setting of the cement paste containing natural fibers can be attributed to the ionization of pectin from the natural fibers, owing to the alkaline pore solution formed during cement hydration. This pectin binds with calcium ions in the cement matrix, reducing its concentration and thereby delaying setting [24].

As shown in Figure 6b, the cumulative heat release results further illustrate this trend. For all specimens, the cumulative heat release increased almost linearly until approximately 12 h, and the cumulative heat release at 12 h was similar for all specimens. For the Plain, NFA, and NFJ specimens, the cumulative heat release increased rapidly after 12 h, whereas for the NFH specimen the increase was delayed until after 16.7 h. After 48 h, the cumulative heat release of the Plain specimen was the highest; however, beyond 48 h, the cumulative heat release for the NFA and NFJ specimens exceeded that of the Plain specimen. For the NFH specimen, the cumulative heat release exceeded that of the Plain after approximately 64 h. During the initial hydration, the inclusion of natural fibers inhibited the hydration reaction of the cement, resulting in a lower cumulative heat release compared with the Plain specimen. However, after a certain period, the cumulative heat release of the specimens containing natural fibers exceeded that of the Plain specimen. This is believed to be due to the decomposition of the components of natural fibers under the elevated pH resulting from the initial cement hydration, which initially inhibited the hydration reaction, followed by the natural fibers providing additional nucleation sites for cement hydration products, thereby promoting the hydration reaction [43].

### 3.3. Compressive-Strength Results

Figure 7 illustrates the compressive-strength results of the mortar specimens containing natural fibers at different curing ages (7, 28, and 91 d). The increasing trend reflects the typical progression of the compressive-strength development owing to the ongoing hydration process, which leads to the formation of hydration products and densification of the pore structure. The specimens incorporating natural fibers generally exhibited lower compressive strengths than the control specimen. These findings suggest that the incorporation of natural fibers can reduce the compressive strength of the mortar at early curing ages; however, the compressive strength does improve substantially with curing age, converging on that of the Plain specimen after 90 d.

The reduction in the compressive strength at early curing ages can be explained by two primary factors. First, the inhibition of cement hydration was due to the chemical components present in the natural fibers. As discussed in Section 3.1 and Section 3.2, these components decompose in the alkaline environment of the cement matrix, leading to a reduction in the calcium ion concentration and thereby suppressing the hydration reaction of the cement clinker [42]. Second, the inherent porosity of natural fibers and their weak bonding with the cement matrix play significant roles. The lumen structure of the natural fibers contributes to an increase in the overall porosity of the cement matrix, which in turn reduces the compressive strength. Additionally, the weak adhesion between the natural fibers and cement matrix reduces the continuity of the matrix and increases its porosity, further lowering the strength.

However, as the curing age increased, the bond strength between the natural fibers and the cement matrix improved, and the hydration products gradually filled the areas surrounding the fibers, leading to a recovery in the compressive strength. This phenomenon can be attributed to the internal curing effect resulting from the high water-absorption capacity of the natural fibers [24,29,30]. Natural fibers have a high capacity for water absorption, owing to their porous structure. They absorb a part of the mixed water from the cement paste and store it internally, gradually releasing the stored water as hydration progresses, thereby providing an additional internal source of moisture within the cement matrix [24]. This released water promoted further hydration of the cement particles, thereby increasing the degree of hydration of the unreacted cement. This effect is particularly beneficial in deep regions, where the penetration of external curing water is limited. Additionally, the hydration products such as C-S-H gel formed during further hydration fill the pores around the fibers, increasing the density of the matrix and reducing its porosity. Consequently, the internal curing effect led to an increase in the compressive strength at later curing ages. Figure 7 suggests that the abaca and jute fibers may interact more effectively with the cement matrix than the hemp fibers, and indicating that hemp fibers may have relatively lower water absorption and weaker bonding with the cement matrix because of their chemical characteristics, compared with other natural fibers.

Figure 8 illustrates the increase in the compressive strength for each specimen from 7 to 28 d, and from 28 to 91 d. This trend for the Plain specimen reflects the typical behavior of cement paste, where hydration reactions proceed actively at early curing ages, resulting in significant strength development. However, the rate of increase diminishes over time, as hydration slows. Specimens incorporating natural fibers exhibited slightly lower compressive-strength gains than the Plain specimen during the early curing period (7–28 d). This reduction in strength gain could be attributed to the inhibitory effect of natural fibers on early hydration or the absorption of mixed water by the fibers, which may have temporarily reduced the water supply in the cement matrix.

However, during the longer curing period, from 28 to 91 d, the strength gains for the specimens with natural fibers were greater than those for the Plain specimen. NFA and NFJ specimens exhibited pronounced internal curing effects, which were significantly higher compared to that for the Plain specimen. This suggests that the internal curing effect of the high water-absorption capacity of natural fibers becomes more prominent at longer curing ages. In contrast, the NFH specimen exhibited a strength increase that was somewhat less than that of the other fiber-reinforced specimens. The incorporation of natural fibers may slightly inhibit the early development of compressive strength; however, it can significantly enhance the strength at longer curing ages, owing to the internal curing effect. This is because the high water-absorption capacity of natural fibers promotes prolonged hydration reactions, improves the microstructure, and enhances the mechanical properties of the material.

Figure 9 presents the SEM images of the mortar specimens containing natural fibers after 91 d of curing. Figure 9a shows the interfacial zone between the natural fibers and the cement matrix. The absence of visible pores or gaps at the interface indicated that the internal curing effect of the natural fibers facilitated additional hydration, resulting in the formation of hydration products that densified the matrix structure. Figure 9b shows the natural fibers embedded in the fractured surface of the mortar specimen. As shown in Figure 9b, the surfaces of the natural fibers were coated with cement-hydration products. This interaction between the natural fibers and cement matrix contributes to the densification of the microstructure at longer curing ages, thereby improving the compressive strength and other mechanical properties.

### 3.4. Microstructural Analysis Using MIP

Figure 10 presents the results of MIP conducted after seven days of curing, showing the log differential intrusion and cumulative intrusion data for the specimens. As shown in Figure 10a, the primary micropores in the specimens at seven days were in the range of 50–70 nm in size. For the specimens incorporating natural fibers, the pore sizes were larger than those of the Plain specimen. From the results in Figure 10b, the cumulative intrusion for the Plain specimen at seven days was 0.126 cm^3^/g. In contrast, all the specimens incorporating natural fibers exhibited an increase in cumulative intrusion compared to the Plain specimen. Specifically, the NFA, NFJ, and NFH specimens showed 0.142, 0.146, and 0.149 cm^3^/g, respectively, representing increases of approximately 12.7, 15.9, and 18.3%, respectively, compared to the Plain specimen. This indicates that the incorporation of natural fibers increased the porosity of the cement paste. In Figure 10b, the higher cumulative intrusion observed at pore diameters exceeding 100 nm in the specimens containing natural fibers can be attributed to the porous lumen structure inherent to the natural fibers themselves.

An increase in porosity within the cement matrix implies an increase in the total pore volume, which significantly affects the physical and mechanical properties of the material. The increased porosity can be attributed to several factors owing to the incorporation of natural fibers. The uneven distribution of natural fibers can lead to the formation of small gaps at the interface between the fibers and the cement matrix. Additionally, the inherently porous structure of natural fibers contributes to additional pore formation within the cement matrix. Increased porosity generally leads to a decrease in the compressive strength, which aligns with the compressive-strength results shown in Figure 7.

Figure 11 shows the results of MIP conducted after 91 d of curing, illustrating the log-differential intrusion and cumulative intrusion of the specimens. In contrast to the results shown in Figure 10a, Figure 11a shows that the primary pore size of the specimens incorporating natural fibers decreased as the curing age increased. For the NFA, NFH, and NFJ specimens, the primary pore size decreased to approximately 50 nm, like that of the Plain specimen, indicating a reduction in pore size with increasing curing age. As shown in Figure 11b, the cumulative intrusion of the Plain specimen at 91 d was 0.096 cm^3^/g, which was a significant decrease from 0.126 cm^3^/g at 7 d. This reduction suggests that the hydration process of the cement paste led to the densification of the pore structure and a reduction in the total pore volume over time. For the specimens incorporating natural fibers, the cumulative intrusion at 91 d showed a more pronounced reduction than that of the Plain specimen. The cumulative intrusion values for NFA, NFJ, and NFH were 0.101, 0.100, and 0.114 cm^3^/g, respectively. Although the NFA and NFJ specimens had cumulative intrusion values similar to those of the Plain specimen, the NFH specimen maintained relatively higher cumulative intrusion.

The significant decrease in the cumulative intrusion of the fiber-reinforced specimens at 91 d can be attributed to the internal curing effect caused by the high water-absorption capacity of the natural fibers. The water stored within the natural fibers is gradually released into the cement matrix, increasing the hydration degree of the unreacted cement particles and filling the pores with hydration products, thereby enhancing the matrix density and reducing the porosity. This reduction in porosity owing to internal curing in the fiber-reinforced specimens correlates with the substantial increase in compressive strength observed at 91 d. The decrease in porosity contributed to an increase in the matrix density and an improvement in the microstructure, ultimately enhancing the mechanical properties of the material.

## 4. Conclusions

This study experimentally investigated the effects of natural fibers on the hydration and micropore structure of cement matrices. The main findings are summarized as follows:All natural fibers tested—abaca, hemp, and jute fibers—delayed the initial hydration reaction of the cement paste, resulting in an increase in the setting time. The final setting time increased significantly compared with the initial setting time, when natural fibers were incorporated. The hemp fibers caused a more pronounced increase in the final setting time than the abaca and jute fibers. This retardation in cement hydration is attributed to the chemical components of natural fibers such as cellulose, pectin, and lignin, which leach into the pore solution of the cement, inhibiting the hydration reaction.Cement composites incorporating natural fibers exhibited lower compressive strengths at an earlier curing age (seven days) than the Plain specimen. This was due to the inhibition of early hydration reactions caused by fiber incorporation and the increase in porosity from the fibrous porous structure. However, as the curing age increased, the internal curing effects caused by the gradual release of water stored within the natural fibers led to a significant increase in compressive strength. The specimens with abaca and jute fibers reached similar strength levels as the Plain specimen at 91 d, whereas the specimens with hemp fibers maintained a slightly lower compressive strength.After seven days of curing, the cement composites with natural fibers exhibited higher porosity than the Plain specimen, which was attributed to the porous structure of the fibers and the interfacial gaps between the fibers and cement matrix. At a longer curing age (91 d), the internal curing effects resulted in a reduction in the porosity and pore size, leading to the densification of the cement matrix.The high water-absorption capacity of natural fibers allows them to store a portion of the mixing water and gradually release it during hydration, providing an additional internal source of moisture within the cement matrix. This internal curing effect promotes the formation of hydration products in the cement composite at longer curing ages, increasing the degree of hydration of the unreacted cement, and thereby contributing to strength recovery and microstructural improvement.Cement composites with natural fibers have demonstrated their potential as sustainable construction materials. Specifically, the internal curing effect allows for a uniform and continuous moisture supply, even in large structures or environments in which external curing is challenging, leading to improved long-term durability and performance. This could contribute to the development of eco-friendly and cost-effective building materials.To maximize the compressive strength and improve the microstructure of natural fiber-reinforced cement composites, further studies on the chemical or thermal treatment of fibers, optimal incorporation ratios, and effective mixing methods are required. Additionally, long-term studies that evaluate the practical performance of natural fiber-based composites, including their durability and creep, are required for further development.

## Figures and Tables

**Figure 1 materials-18-00059-f001:**
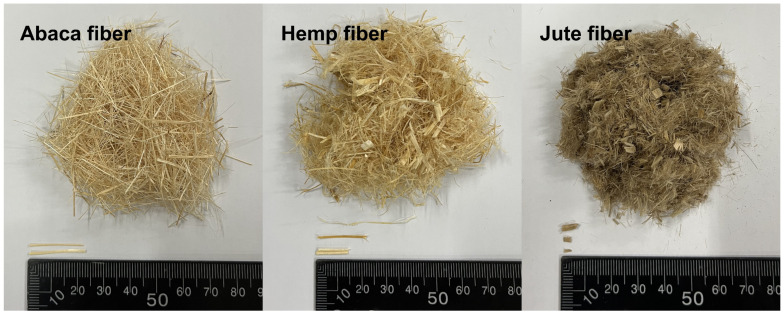
Optical images of natural fibers used.

**Figure 2 materials-18-00059-f002:**
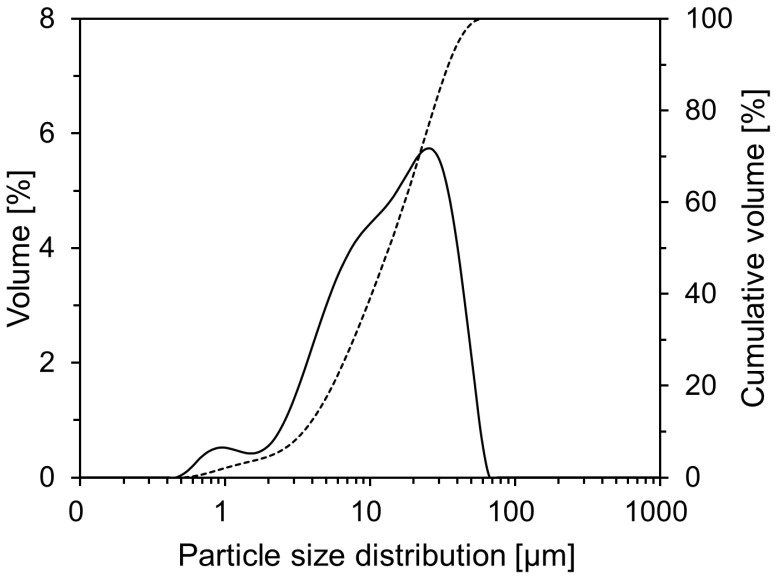
Particle size distribution of OPC.

**Figure 3 materials-18-00059-f003:**
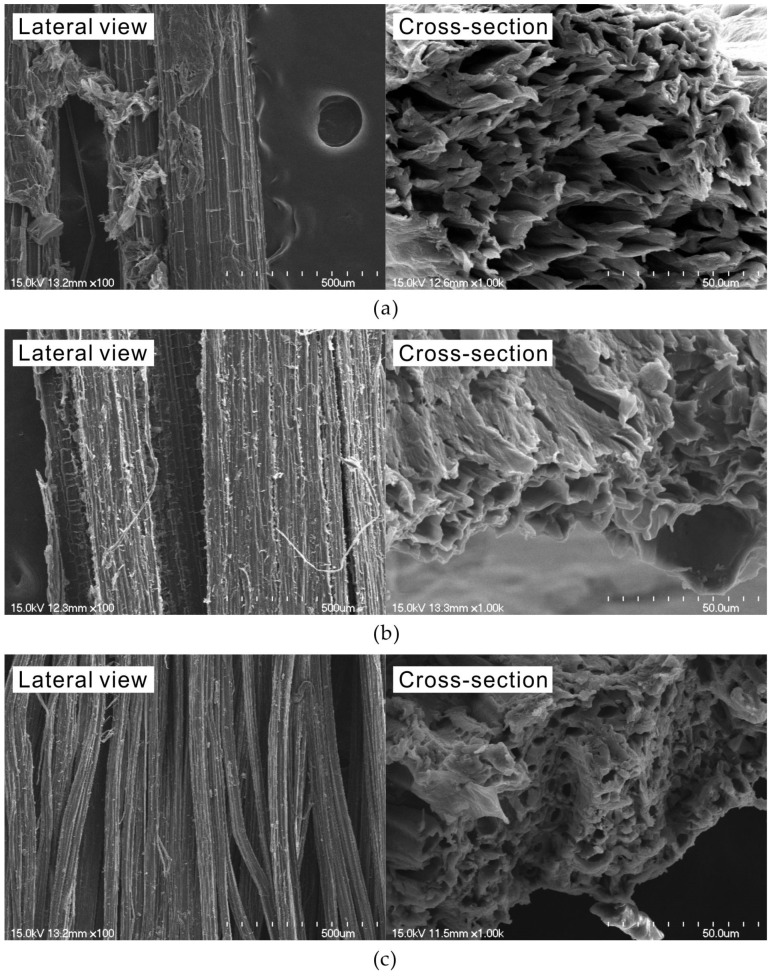
SEM images of natural fibers: (**a**) abaca; (**b**) hemp; (**c**) jute.

**Figure 4 materials-18-00059-f004:**
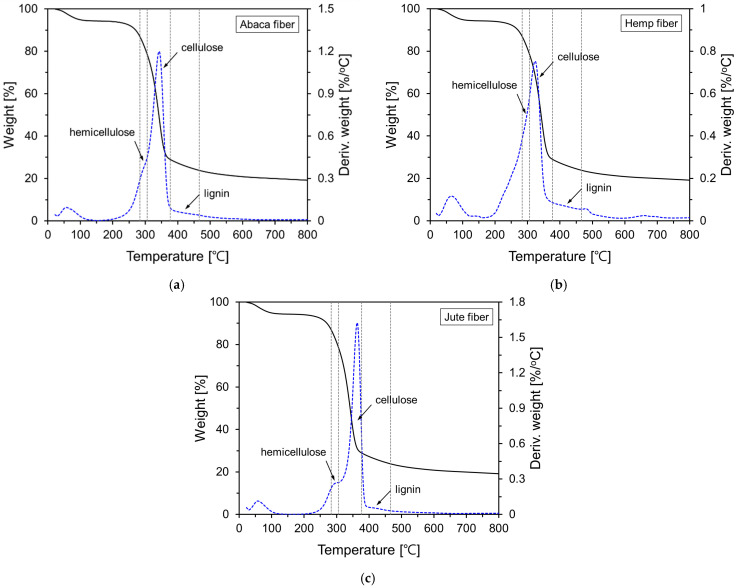
TG and DTG analysis results of natural fibers: (**a**) abaca; (**b**) hemp; (**c**) jute.

**Figure 5 materials-18-00059-f005:**
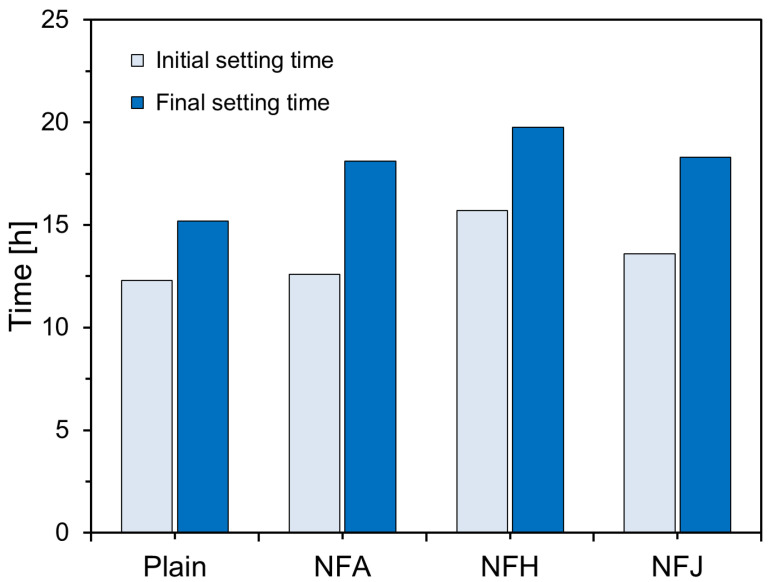
Setting times of specimens.

**Figure 6 materials-18-00059-f006:**
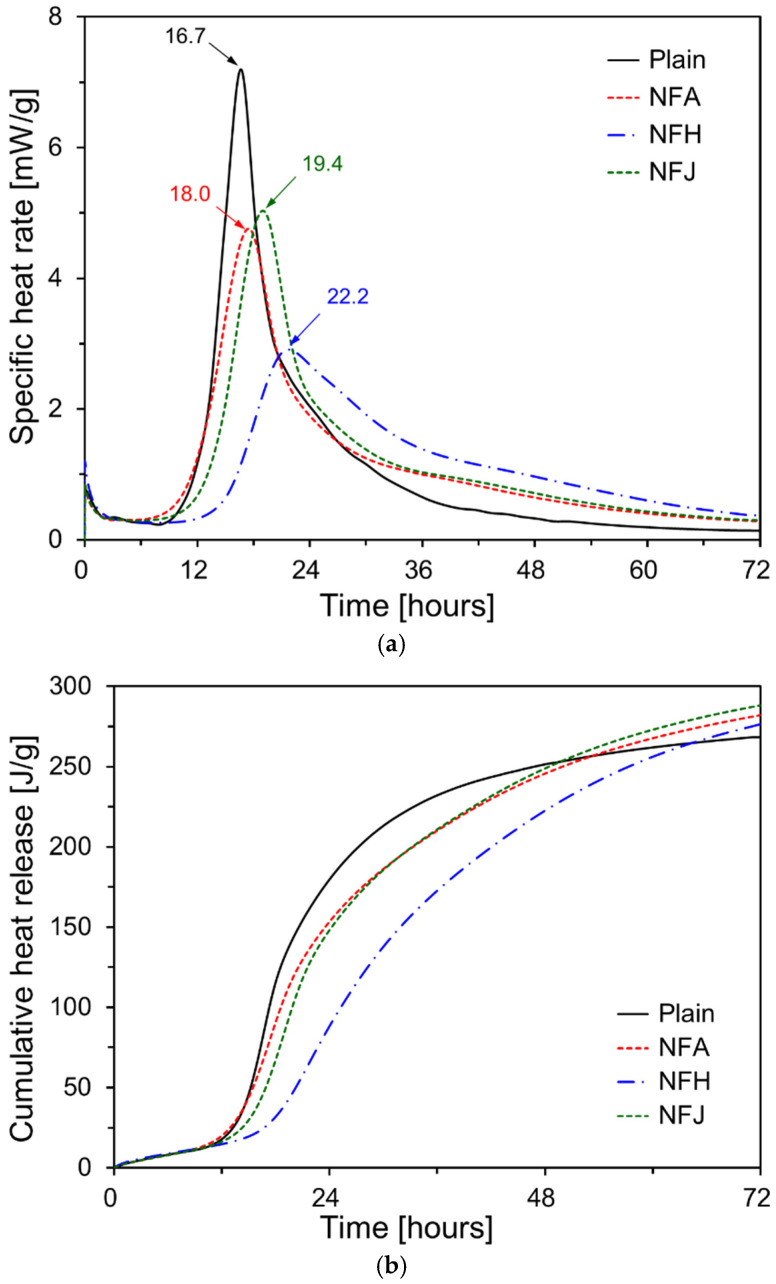
Heat of hydration results: (**a**) specific heat rate; (**b**) cumulative heat release.

**Figure 7 materials-18-00059-f007:**
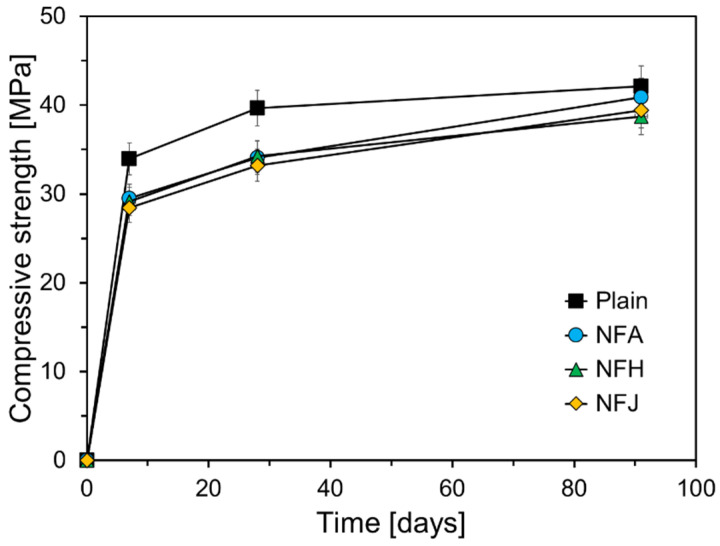
Compressive-strength results.

**Figure 8 materials-18-00059-f008:**
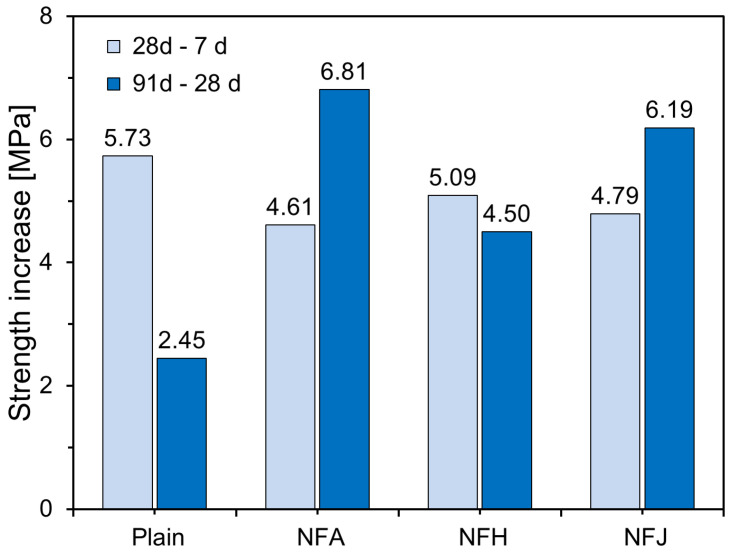
Compressive-strength increases of specimens.

**Figure 9 materials-18-00059-f009:**
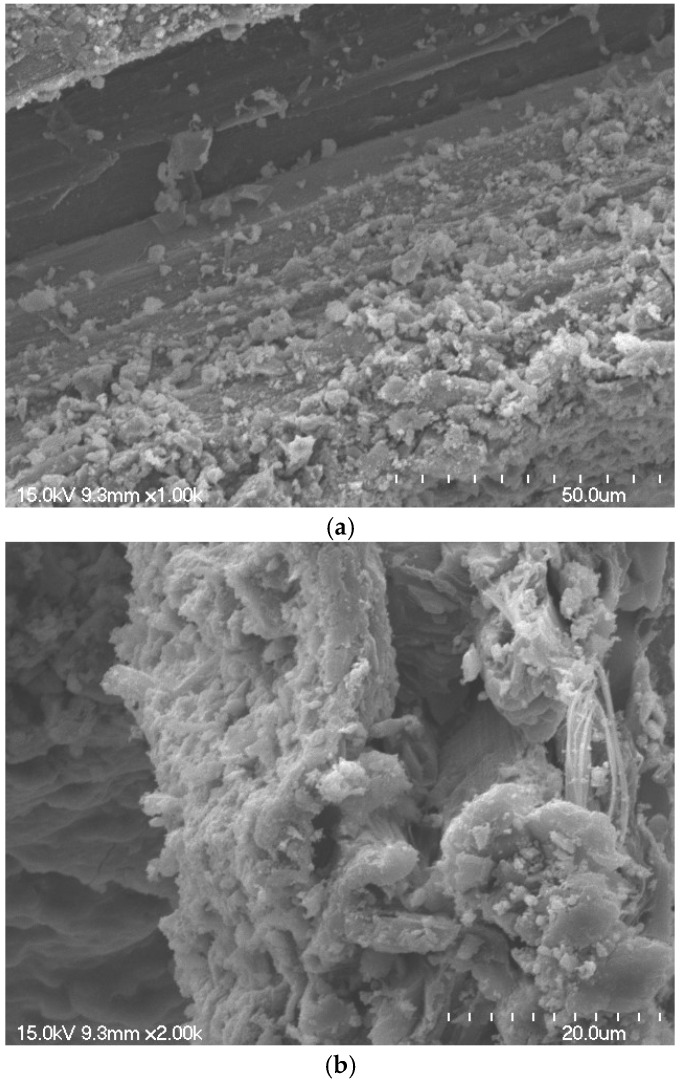
SEM images of mortar specimen: (**a**) ×1000; (**b**) ×2000.

**Figure 10 materials-18-00059-f010:**
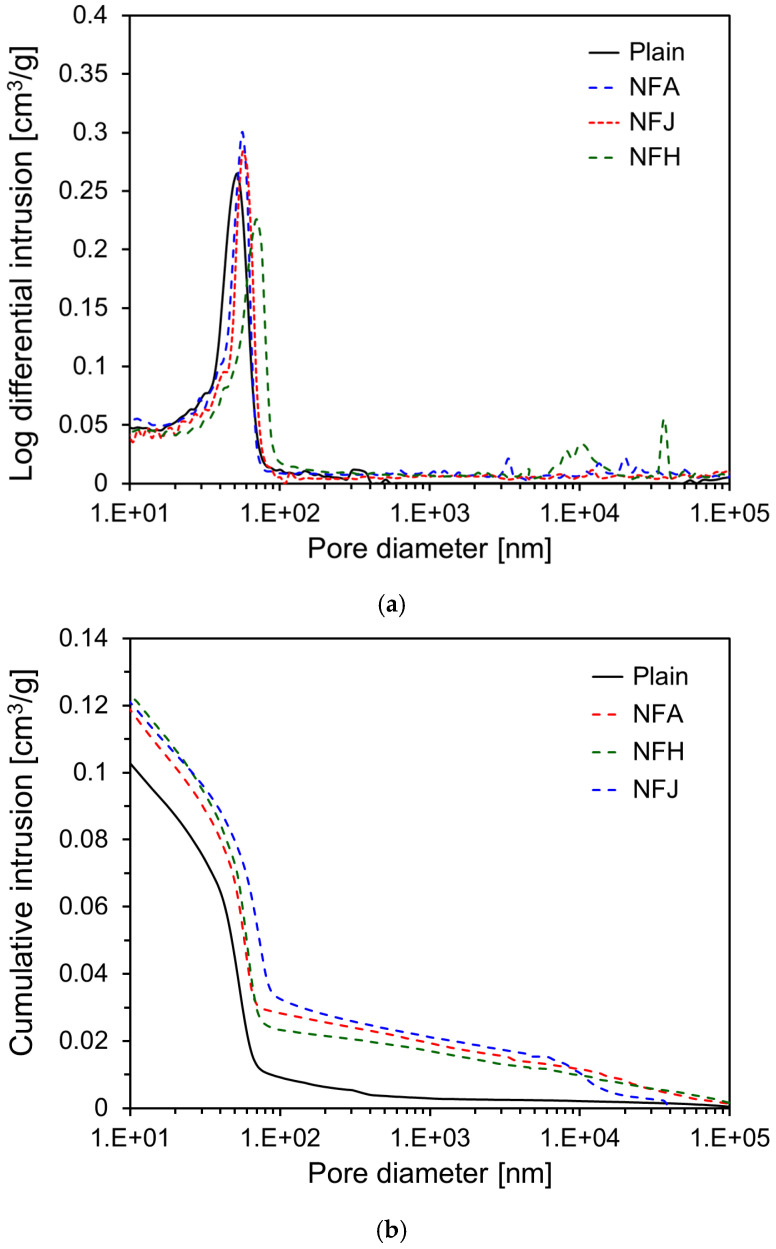
MIP test results of specimens at seven days: (**a**) log differential intrusion; (**b**) cumulative intrusion.

**Figure 11 materials-18-00059-f011:**
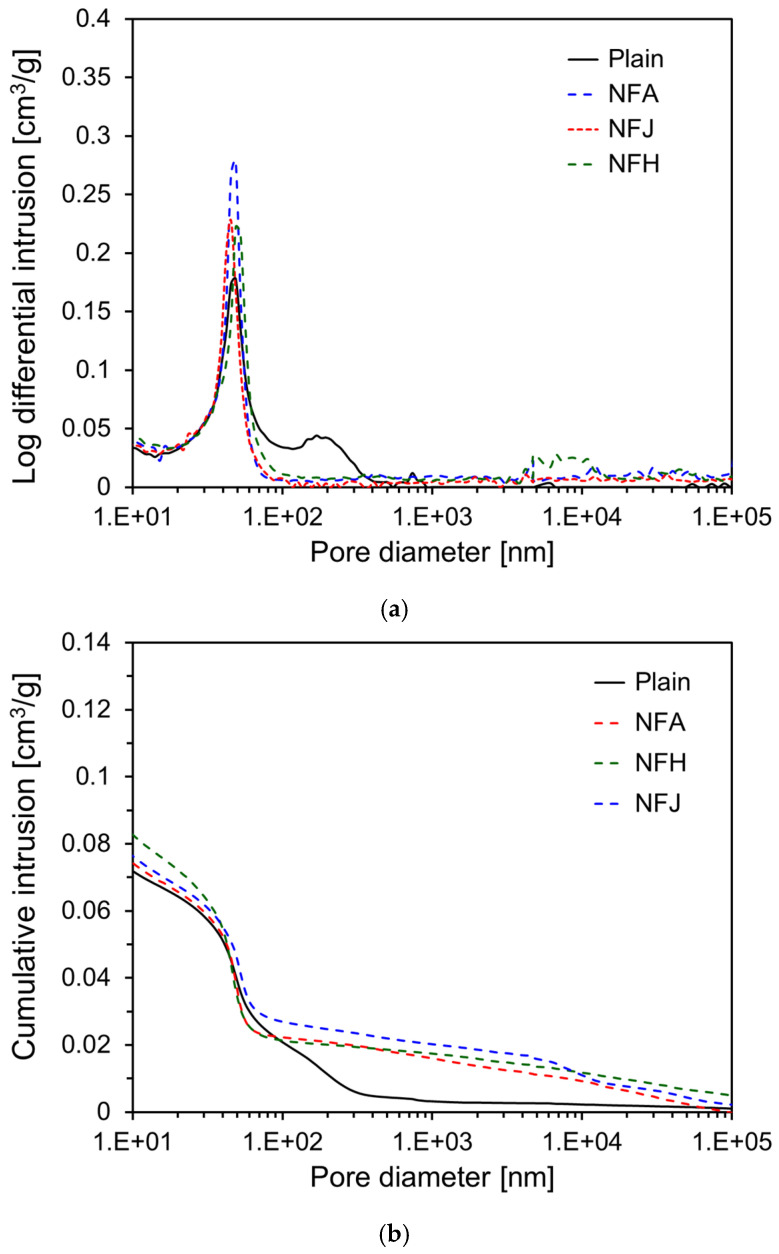
MIP test results of specimens at 91 d: (**a**) log differential intrusion; (**b**) cumulative intrusion.

**Table 1 materials-18-00059-t001:** Chemical oxide compositions of OPC.

	Chemical Oxide Compositions (wt%)								
	SiO_2_	Al_2_O_3_	Fe_2_O_3_	CaO	MgO	K_2_O	Na_2_O	SO_3_	LOI
OPC	21.8	5.3	3.7	59.8	2.6	1.1	0.7	2.5	1.2

**Table 2 materials-18-00059-t002:** Physical properties of natural fibers.

	Density (g/cm^3^)	Tensile Strength (MPa)	Elastic Modulus (GPa)
Abaca fiber [32]	1.50	430–760	12
Hemp fiber [33,34]	1.48	500–900	70
Jute fiber [34]	1.44	392–800	20

**Table 3 materials-18-00059-t003:** Chemical compositions of natural fibers [37].

	Cellulose (wt%)	Hemicellulose (wt%)	Lignin (wt%)
Abaca fiber	56.00–68.32	15.0–30.7	7.0–15.1
Hemp fiber	70.2–74.4	17.9–22.4	3.7–5.7
Jute fiber	61.0–71.5	13.6–20.4	12–13

**Table 4 materials-18-00059-t004:** Mixture proportions of cement composites.

Specimens	Cement (kg/m^3^)	Water (kg/m^3^)	Extra Water (kg/m^3^)	Fiber Contents (wt% by Cement)		
				Abaca	Hemp	Jute
Plain	202	672	-	-	-	-
NFA	202	672	19.7	1	-	-
NFH	202	672	14.4	-	1	-
NFJ	202	672	18.3	-	-	1

## Data Availability

The original contributions presented in this study are included in the article. Further inquiries can be directed to the corresponding author.

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
