# Peer review of "Microstructural Characteristics of Cellulosic Fiber-Reinforced Cement Composite"

_materials, 2024, doi:10.3390/ma18010059_

Round 1

Reviewer 1 Report

Comments and Suggestions for Authors

 This study employed advanced techniques like mercury intrusion porosity and isothermal calorimetry to uncover the influence of natural fibers such as abaca, hemp, and jute on the hydration behaviour and pore structure of cement composite. The manuscript is insightful. To further enhance its impact and contribution to the field, I recommend considering the following refinements:

1- The manuscript is well-written, and the author's English language skills are excellent.

2-The abstract provides a good overview, but it could be more concise to better capture the main findings.

3-The introduction could benefit from a more comprehensive literature review. Relevant previous studies should be discussed to justify the research.

4- The authors do not provide adequate information on the specific properties of the natural fibers used, such as their tensile strength. This information could help clarify how the different fibers influence the results.

5-There is a lack of information about the sand used to prepare the mortar specimens.

6-The results are clearly presented; however, the discussion could use a deeper interpretation of the findings.

7- The application of advanced techniques to assess microstructural changes and hydration characteristics improved the reliability of the results.

8- The findings highlight the potential of natural fibers to improve the long-term performance of cement composites. This insight is significant for academic research and practical applications in the construction sector.

9- The author should ensure that all references are consistently formatted according to the journal guide.

 The manuscript contributes to the broader goal of developing eco-friendly construction materials. The reviewer recommends publication upon addressing the noted points.

Comments on the Quality of English Language

Can be improved.

Author Response

Comments : Please see the attachment.

English : The authors have revised the entire manuscript by professional editing service (Editage, www.editage.co.kr).

Reviewer 2 Report

Comments and Suggestions for Authors

The microstructural and hydration characteristics of cement composites reinforced with abaca fiber, hemp fiber and jute fiber have been studied in this paper. The influences on the hydration behavior and pore structure of cement composite have been evaluated. The results show that the hydration process and microstructural development of the cement composites are affected by the natural fibers. The durability and sustainability can be enhanced potentially. The intial hydration cement is delayed by the natural fibers.

Some specific comments related to the manuscript are as follows.

1. It’s better to increase the literatures about the improvement of mechanical performance of cement matrices by natural fibers.

2. Why did you select the ordinary Portland cement as the materials?

3. What are the differences about the cross-sections of three different natural fibers?

4. How to control the mix proportions for the cement composites?

5. The factors for the reduction in compressive strength should be comfirmed.

Comments on the Quality of English Language

Moderate editing of English language required.

Author Response

(The authors gave the same response as above.)

Reviewer 3 Report

Comments and Suggestions for Authors

This study investigated microstructural and hydration characteristics of cement composites reinforced with three types of natural fibers (abaca fiber, hemp fiber, jute fiber). The fibers were selected to evaluate their influence on the hydration behavior and pore structure of cement composite. 

The test results demonstrated that natural fibers affect both the hydration process and the microstructural development of the cement composites, potentially enhancing durability and sustainability. The test results indicated that natural fibers tend to delay the initial hydration of cement, but over the long term, they enhance compressive strength and densify the microstructure due to their internal curing effect.

The paper could be considered for publication after the revision.

-Why the abaca, hemp and jute fibers were chosen ? Simple wood fibers should be tested here also for comparison.

-The testing of wood fibers in the field should be described in introduction.

-What amount of cellulose, lignin, pectin is in the different fibers ? What is effect of the components for properties of the cement composites ?

-How the fibers were processed before the investigations ?

-There is a weight loss in TGA even at about 60-70 oC. How this could be explained ?

-Fiber content (wt% by cement) was 1. Why other amounts were not tested ?

-The products are mentioned as cost-effective building materials. Could some calculations be provided to demonstrate the cost-effective factor ?

Round 2

Reviewer 2 Report

Comments and Suggestions for Authors

The authors have revised the manuscript according to the comments. This paper can be accepted.

Comments on the Quality of English Language

The English could be improved to more clearly express the research.

Author Response

Comment 1 : The authors have revised the manuscript according to the comments. This paper can be accepted.
Response 1 : Thank you for good comment. 

The authors have revised the entire manuscript by professional editing service (Editage, www.editage.co.kr).

Reviewer 3 Report

Comments and Suggestions for Authors

There is a weight loss in TGA even at about 60-70oC. How this could be explained ?

Answer of author is not suitable. This should be explained correctly. OH groups can not be removed from materials at this temperature ?
